# Comparison of Physicochemical Characteristics and Macrophage Immunostimulatory Activities of Polysaccharides from *Chlamys farreri*

**DOI:** 10.3390/md18080429

**Published:** 2020-08-17

**Authors:** Fulin Shi, Zhicong Liu, Yang Liu, Kit-Leong Cheong, Bo Teng, Bilal Muhammad Khan

**Affiliations:** Department of Biology & Guangdong Provincial Key Laboratory of Marine Biotechnology, Institute of Marine Sciences, College of Science, Shantou University, Shantou, Guangdong 515063, China; 16flshi@alumni.stu.edu.cn (F.S.); 17zcliu2@stu.edu.cn (Z.L.); klcheong@stu.edu.cn (K.-L.C.); bteng@stu.edu.cn (B.T.); khan@stu.edu.cn (B.M.K.)

**Keywords:** *Chlamys farreri* polysaccharides, physicochemical analysis, RAW264.7 cells, immunostimulatory activity

## Abstract

To address the structure–activity relationship of *Chlamys farreri* polysaccharides on their immunostimulatory efficacy, two polysaccharides (CFP-1 and CFP-2) were extracted from *Chlamys farreri* by hot water extraction, and separated through column chromatography. The isolated CFPs were chemically analyzed to clarify their physicochemical characteristics and cultured with murine macrophage RAW264.7 cells, in order to evaluate their immunostimulatory efficacy. Despite the fact that both CFP-1 and CFP-2 were mainly comprised of glucose lacking the triple-helix structure, as revealed through preliminary physicochemical analyses, obvious differences in regard to molecular weight (Mw), glucuronic acid content (GAc) and branching degree (BD) were observed between CFP-1 and CFP-2. In in vitro immunostimulatory assays for macrophage RAW264.7 cells, it was demonstrated that CFP-2 with larger Mw, more GAc and BD could evidently promote phagocytosis and increase the production of NO, IL-6, TNF-α and IL-1β secretion, by activating the expression of iNOS, IL-6, TNF-α and IL-1β genes, respectively. Hence, CFP-2 shows great promise as a potential immunostimulatory agent in the functional foods and nutraceutical industry, while CFP-1, with lower molecular weight, less GAc and BD, displays its weaker immunostimulatory efficacy, based on the indistinctive immunostimulatory parameters of CFP-1.

## 1. Introduction

A multitude of monosaccharide units bonded through glycosidic linkages constitute polysaccharides, which frequently exist in plants, bacteria, fungi and animals, as a vital biological macromolecule. The low toxicity associated with polysaccharides, together with their wide-ranging bioactivities, like antidiabetic [1], anti-inflammatory [2,3], antitumor [4], antioxidation [5,6], antiobesity [7] and immunomodulation [8,9] have contributed towards their increasing popularity in the scientific community. Similarly, immunostimulatory polysaccharides can directly or indirectly activate the immune system by triggering several cellular or molecular events [10]. The chief mediators of the action of such polysaccharides are reported to be monocytes, macrophages, and neutrophils [11], with most studies focusing on the function of macrophages [12]. 

The study about the polysaccharide from *Sarcodon aspratus*, for instance, showed that its immunostimulatory activity not only enhanced the phagocytic function of RAW264.7 cells, but also increased the production of nitric oxide (NO), reactive oxygen species (ROS), and cytokines (TNF-α and IL-6) [13]. Similar results were also found in a low-molecular-weight β-glucan from *Durvillaea Antarctica*, which could significantly activate RAW264.7 cells to release NO, ROS, TNF-α, MCP-1 and IL-1β, and exerted stronger immunostimulatory activity on RAW264.7 cells compared with lentinan, in many aspects [14]. Moreover, some studies reported that immunostimulatory polysaccharides affected the proliferation and differentiation of macrophages as well [15,16]. With the developments in polysaccharides research, it has been found that the biological activities of polysaccharides are closely related to their structure. The structure-activity relationship of polysaccharides, hence, has become an essential part of polysaccharide research.

Marine invertebrates are attracting the attention of researchers in recent years, due to their nutritive value, potential health benefits and therapeutic applications [17]. They are rich in protein, amino acids, carbohydrate, vitamins, and inorganic elements. The active proteins isolated from these invertebrates have been reported to have antioxidant, antifatigue and immunostimulatory activity [18]. In addition to proteins, some bioactive polysaccharides have been discovered, such as the Jellyfish skin polysaccharides, which showed strong inhibitory effects on oxidized low-density (oxLDL) induced conversion of macrophages into foam cells [19]. Likewise, the sulfated polysaccharides obtained from sea cucumber had stronger anticoagulant activity [20].

*Chlamys farreri* is a member of the *Pectinidae* family, which is naturally distributed throughout the coasts of East Asia, and is a commercially available mollusk in China [21]. Various bioactivities, including antioxidant, antitumor and antiviral activities, have been reported for the polysaccharides in shellfish [22], however, its immunostimulatory efficacy remains unexplored. In this study, the crude polysaccharide isolated from the *Chlamys farreri* (CFP) by hot water extraction was separated into two fractions (CFP-1 and CFP-2), through DEAE-52 cellulose chromatography, while these fractions were homogenized using Sephadex G75 chromatography. These fractions were then subjected to physicochemical analyses, including molecular weight, monosaccharide composition and structural features. Moreover, the immunostimulatory activity of crude CFP, CFP-1 and CFP-2 were investigated in vitro, using murine macrophage RAW 264.7 cells. In addition, the molecular level study of their immunomodulation efficacy was performed using RT-QPCR.

## 2. Results and Discussion

### 2.1. Isolation and Purification of CFPs

#### 2.1.1. Optimization of Extraction Conditions

The single-factors tests were performed, to determine the effects of different factors on the extraction efficiency of CFP (Table 1). The total sugar yield increased from 26.31% to 28.78%, as the solvent to material ratio (V/W) changed from 30 to 60. A change in this ratio from 60 to 70, however, did not exhibit any obvious differences in this regard. Likewise, a rapid increase in the yield was detected up to 65 °C, demonstrating a direct proportionality between high temperature and increased yield. A slight increase in yield was also observed at 95 °C. Similar patterns of extraction were also reported in other polysaccharides’ extraction [19]. In a similar fashion, the yield improved, with an increase in extraction time from 1 to 4 h, while no significant differences in this regard were observed upon increasing the extraction time up to 5 h. The extraction rate under optimum conditions (solvent to material ratio of 60, 65 °C and 4 h) was 29.84%.

#### 2.1.2. Removal of Proteins from Crude Polysaccharides

Polysaccharides are traditionally deproteinized, either by denaturing the proteins by chemical reagents, or through their enzymatic hydrolysis with the help of proteases [23]. A more environment friendly and convenient method, ethanol-ammonium sulfate ATPS, has also been reported as an effective tool in this context [4,5,24]. In the present study, sevag method exhibited a higher residue rate of protein (39.81 ± 4.10%) and a higher loss rate of polysaccharide (18.28 ± 2.37%) than the ethanol-ammonium sulfate ATPS method (Table 2). In addition, Sevag regent contains poisonous chloroform, which is environmentally disadvantageous. It is for this reason that ethanol-ammonium sulfate ATPS was preferentially chosen for the removal of proteins from CFP.

#### 2.1.3. Separation of Deproteinized Polysaccharide

The crude CFP was collected by the ethanol precipitation method, and then purified by DEAE cellulose-52 column chromatography. The HPLC chromatogram represented two observable peaks (Figure 1A). The peak representing the neutral polysaccharide was eluted with distilled water, while 0.3 mol/L NaCl was used as an eluent for the acidic polysaccharide. After being collected, the polysaccharides were further purified by Sephadex G-75 with pure water as eluent (Figure 1B,C). The main fractions (CFP-1 and CFP-2), as depicted by the elution curve, were collected and concentrated at 50 °C before subjecting them to freeze-drying. The freeze-dried of the crude CFP, purified CFP-1 and CFP-2 will be for further research.

### 2.2. Chemical-Physical Properties of Polysaccharides

#### 2.2.1. The Molecular Weight of CFP-1 and CFP-2

Both CFP-1 and CFP-2 exhibited a single and symmetric peak in the HPGPC chromatograms, indicating that they are homogeneous polysaccharides (Figure 2). The retention time of CFP-1 was 20.85 min, and that of CFP-2 was 9.80 min. On the basis of the equation derived from the standard curve, the molecular weights of CFP-1 and CFP-2 were calculated to be 8436 Da and 82372 Da, respectively.

#### 2.2.2. Chemical Composition Analysis

The glucuronic acid content of crude CFP, CFP-1 and CFP-2 was measured by the m-hydroxydiphenyl method, and calculated as 12.01% ± 2.051%, 3.73% ± 1.04% and 24.09% ± 5.03%, respectively. For the monosaccharide composition analysis, the results from HPLC spectrum (Figure 3B) indicated that crude CFP was composed of glucuronic acid, galacturonic acid, and glucose, with a molar ratio of 1:1.92:29.13. HPLC only detected traces of galacturonic acid and glucuronic acid, mainly because the hydrolysis time is too long, the high temperature of 120 °C, and also the strong oxidizing effect of TFA, so the two uronic acids may be destroyed; it would affect their quantification. However, if the hydrolysis time was shortened, it may also lead to the crude CFP, which cannot be quantified accurately by incomplete hydrolysis. These results indicated that glucose is the main sugar unit, and because the content of these two uronic acids was very low, it did not affect the judgment of the main glycosidic linkage.

#### 2.2.3. Fourier Transform Infrared (FT-IR) Spectrum and UV Scanning Spectrum of CFP-1 and CFP-2

FT-IR spectrum analysis is an essential method to explore the structure of polysaccharides, because of the characteristic absorption of each functional group [25]. As shown in Figure 4, the broad stretching peaks in the region of 3385 cm^−1^ (CFP-1) and 3404 cm^−1^ (CFP-2) were assigned to the hydroxyl stretching vibration of the polysaccharide, and the absorption peak at 2938 cm^−1^ (CFP-1) and 2937 cm^−1^ (CFP-2) designated C-H stretching vibration [26]. The absorbance of COO- deprotonated carboxylic group or bound water was indicated by the absorption peaks at 1624 cm^−1^ (CFP-1) and 1623 cm^−1^ (CFP-2) [27]. Moreover, the broad stretching peak in the region of 950 cm-1 ~ 1200 cm-1 was indicative of ring vibrations overlapping with the stretching vibrations of C-OH side group and the (C-O-C) glycosidic band vibration [28]. The absorptions at 1021 cm^−1^, 1079 cm^−1^, 1155 cm^−1^ (CFP-1) and 1021 cm^−1^, 1079 cm^−1^, 1155 cm^−1^ (CFP-2) testified that the polysaccharides are a pyranose form of carbohydrate [29]. The whole the FT-IR spectra revealed no significant differences between CFP-1 and CFP-2, in regard to the characteristic absorptions of the functional groups (Figure 4).

The UV scanning spectra for both CFP-1 and CFP-2 were void of any absorption at 260 and 280 nm, which was indicative of the absence of proteins and nucleic acids in both fractions (Figure 5).

#### 2.2.4. The Congo-Red Testing for Crude CFP, CFP-1 and CFP-2

The complexation formed by triple-helical polysaccharides and Conge-red can cause the red-shift of maximum absorption in Congo-red solution, and if the triple-helix structure of polysaccharides was destroyed by chemicals, the maximum absorption in Congo-red solution would decrease [25]. 

Curdlan, representing a typical triple-helical polysaccharide conformation, was used as standard in this study. A significant decrease in the maximum absorption wavelength of curdlan and Congo-red complexation at a higher concentration of NaOH (0.5 mol/L) was evident, indicating the destruction of triple-helix conformation (Figure 6). The Congo-red results of crude CFP, CFP-1 and CFP-2 were consistent with that of the standard, wherein the maximum absorption wavelength increased at a NaOH concentration, ranging from 0 to 0.2 mol/L. Conversely, no significant decrease in this regard was observable at higher NaOH concentrations, suggesting that CFPs, unlike curdlan, lacked a triple-helix conformation, and that they may form a new bonding with Congo-red in NaOH solution [25]. Although triple-helix conformation of β-d-glucans has been associated with enhanced immunostimulatory activity in a previous work [30], other reports suggest the contrary. In fact, the immunostimulatory activity of (β1→3)-d-glucans by increasing TNF-α in mouse serum has been reported to be dependent upon the single-helix conformation [31]. Similarly, heteroglucans, lacking a helical conformation, have also been found to possess immunostimulatory function [32,33]. Hence, it is safe to assume that the triple-helix structure is not a necessary factor in determining the immunostimulatory efficacy of polysaccharides.

#### 2.2.5. GC-MS of Alditol Acetate Derivatives from the Methylated Product of CFP-1 and CFP-2

CFP-1 consisted of 1,2-linked Glcp, 1,4-linked Glcp and 3,4,6-linked Glcp, in a molar ratio of 2.12:1.00:6.54, while CFP-2 displayed 1-linked Glcp, 1,4-linked Glcp and 1,3,4-linked Glcp, in a molar ratio of 1.00:9.73:1.72 (Table 3). The DB value for CFP-1 and CFP-2 turned out to be 0.68 and 0.22, respectively. As is known, structure is closely associated with the function of a biological molecule, and branching degree, representing the presence of linked monosaccharides or linked chains, is an important feature of the structure of polysaccharides. For instance, an enhanced immunostimulatory activity has been reported for the highly branched β-(1→3)-d-glucans, in comparison to their less branched or linear counterparts [34,35]. Likewise, more branches of residue units or side chains of fucose, galactose, and/or mannose have been associated with improved immunostimulatory activity in α-d-heteroglucans and β-d-heteroglucans [36,37]. On the other hand, α-(1→6)-d-glucans and α-(1→4)-d-glucans have been found to possess immunostimulatory activity in both linear and branched conformations. In fact, a lower branching degree has been suggested as the structural feature responsible for enhanced immunostimulation in α-(1→4)-d-glucans [38]. In a similar fashion, the less branched CFP-2 in the present study demonstrated a stronger immunostimulatory activity than the relatively highly branched CFP-1, pointing towards the possibility that α-(1→4)-d-glucans form the backbone of CFPs.

### 2.3. Immunostimulatory Activity

#### 2.3.1. Effects of Polysaccharides on RAW264.7 Cell Proliferation

A measure of the proliferation of macrophage, an important immune cell in both innate and adaptive immune response, is regarded as a vital index in the context of cellular immunity [39]. In the present study, the effects of CFPs (in a concentration range of 125 to 2000 μg/mL) on RAW264.7 cell proliferation were examined. As is evident, there were no significant differences between the LPS group (control) and the polysaccharides group, indicating the nontoxicity of CFPs (≤2000 μg/mL) towards RAW264.7 cells (Figure 7A).

#### 2.3.2. Effects of Polysaccharides on Phagocytic Activity of RAW264.7 Cells

A distinguished feature of activated macrophages is illustrated by an increase in phagocytosis [40], which in turn demonstrates the activation of the innate immune response. Neutral red assay was employed here, in order to evaluate the effects of CFPs on the phagocytic activity of RAW264.7 cells. The phagocytic activity was significantly enhanced by LPS, crude CFP and CFP-2, while CFP-1 failed to produce any demonstratable effects in this regard (Figure 7B). In the case of crude CFP, the phagocytic OD values of RAW264.7 increased in a dose-dependent manner, until the concentration was 1000 μg/mL. A slight decline, however, was observed at 2000 μg/mL, indicating that 1000 μg/mL of crude CFP was the optimal concentration for enhancing the phagocytic activity of RAW264.7 cells. A similar trend in this context was also noted for CFP-2. Such an enhancement of phagocytic activity suggests that both crude CFP and CFP-2 are capable of inducing macrophage activation.

#### 2.3.3. Effects of Polysaccharides on NO Production

As an important biological messenger and functional molecule, NO is associated with the physiological function in immune and nervous systems [41]. Activated macrophages initiate an innate immune response to kill the foreign bodies directly by phagocytosis and the release of NO. Consequently, the production of NO is considered as an important index to reflect upon the level of immune activity. In the present study, LPS, crude CFP and CFP-2 were found to be potent in significantly enhancing the production of NO by RAW264.7 cells (Figure 7C). Such an increase in NO production suggested that crude CFP and CFP-2 may activate the bactericidal and tumoricidal activity of macrophages, by binding specific macrophage receptors. Both crude CFP and CFP-2 promoted the release of NO in a dose dependent manner. However, CFP-2 enhanced NO production to a greater extent in comparison to crude CFP. Additionally, NO production was higher in cells treated with CFP-1 (2000 μg/mL) than those in the control group, but this increase was lower than that demonstrated by crude CFP and CFP-2. Furthermore, the increase in NO production by crude CFP and CFP-2 was higher than that reported for Lentinula edodes polysaccharides, wherein the concentration of released NO was less than 4 μmol/L after being treated with 500 μg/mL of the polysaccharide [32]. The results displayed that both crude CFP and CFP-2 had the ability to functionally activate macrophages.

#### 2.3.4. Effects of Polysaccharides on the Cytokines Secretion by RAW264.7 Cells

After being activated, the macrophages can produce a variety of cytokines, such as TNF-α, IL-1β and IL-6, to regulate the cellular and humoral immune responses. As a crucial cytokine, TNF-α not only mediated the immune responses, but also induced inflammatory reactions. CFPs used in this study were able to demonstrate significant effects on the release of TNF-α by RAW264.7 cells (Figure 7D). All tested CFPs enhanced TNF-α levels, but CFP-1 left behind crude CFP and CFP-2 in this regard. Such an enhancement was at its peak at 500 μg/mL and 2000 μg/mL, in the case of crude CFP and CFP-2, respectively, and was significantly higher than that produced by LPS.

IL-1β can regulate the immune response, in addition to being involved in a variety of cellular activities, including the proliferation of T and B lymphocytes [42]. As shown in Figure 7E, LPS, crude CFP and CFP-2 remarkably stimulated the IL-1β secretion by RAW264.7 cells. Crude CFP and CFP-2 enhanced the production of IL-1β in a dose-dependent manner, at concentrations ranging from 125 to 2000 μg/mL, and the effects of CFP-2 were higher than crude CFP at the same concentrations.

IL-6, secreted by immune cells, plays an important role in the regulation of host defense response. A manifest increase in IL-6 secretion was evident for both crude CFP and CFP-2 (Figure 7F). CFP-1, on the other hand, failed to produce any significant effects in this regard.

On the basis of results obtained here in regard to NO, TNF-α, IL-1β and IL-6 production, it is safe to conclude that crude CFP and CFP-2 are capable of functionally activating macrophages, and, hence, can be used as potential immunomodulators. Although both CFP-1 and CFP-2 were fractionated from the same source (crude CFP), the effects of them on RAW264.7 were entirely different. The research data available on polysaccharides suggest that the type, conformation, molecular weight, functional groups and the branching degree of polysaccharides affect their bioactivities, including immunostimulatory efficacy. The apparent differences exhibited by CFP-1 and CFP-2 in stimulating RAW264.7 cells are consistent with their dissimilar physicochemical properties, as previously discussed.

Polysaccharides with an average molecular weight exceeding 1000 Da have been reported to produce immunostimulatory effects in macrophages [43]. Similarly, it has been advocated in previous studies that high molecular weight polysaccharides are more potent in the aspect of immunostimulatory activity than their lower molecular weight counterparts [44]. As is mentioned earlier, the molecular weight of CFP-2 was much higher than that of CFP-1. Besides the molecular weight, the presence of functional groups, like acetyl and sulfate groups, can affect immunostimulatory activity, by changing the charge, solubility and conformation of polysaccharides [45,46]. Mollusks are a rich source of uronic acid-containing polysaccharides (UACPs), possessing biological and pharmacological activities [47]. The glucuronic acid, uronic acid analog of glucose and galactose, the content of CFP-2 was markedly higher than that of CFP-1. A high content of glucuronic acid confers net negative charge on CFP-2. Likewise, the difference in the degree of branching observed for CFP-1 and CFP-2 contributed towards their different immunostimulatory effects, despite both possessing a similar backbone.

#### 2.3.5. Effects of the Polysaccharides on mRNA Expression of iNOS and Cytokines

The transcriptional level effects of CFPs on macrophage activation were determined by measuring iNOS and cytokines (TNF-α, IL-1β and IL-6) mRNA expression in RT-QPCR. Though NO is produced by all three forms of nitric oxide synthase (NOS), inducible (iNOS) is the dominant type of these enzymes during the large production of NO, in the event of tissue injury, cancer tumor suppression and antimicrobial activity [48]. As manifested in Figure 8, the mRNA expression of iNOS (A), TNF-α (B), IL-1β (C) and IL-6 (D) showed a significant increase after treatment with crude CFP and CFP-2. The mRNA expression of iNOS and IL-1β were promoted in a dose-dependent manner. However, CFP-1 showed no effects on mRNA expression of iNOS and the cytokines. These results testified the capability of crude CFP and CFP-2 in enhancing NO, TNF-α, IL-1β and IL-6 secretion, by upregulating their corresponding genes.

#### 2.3.6. Effects of Inhibitors on the Cytokines Secretion by RAW264.7 Cells

As biological macromolecules, polysaccharides mediate immune response through receptors on the surface of cells. Although many receptors were reported to be involved in immune response, TLR4 showed an essential role in the binding of macrophages [49]. Many studies have shown that LPS and polysaccharides can induce NO release and cytokine secretion in activated macrophages, accompanied by the phosphorylation of ERK, JNK and p38 [50]. As shown in Figure 9, it is obvious that theses inhibitors can decrease the secretion of NO (Figure 9A) and TNF-α (Figure 9B) in CFP-2 treated RAW264.7 cells, compared with the CFP-2 only treatment group. These results suggested that CFP can be recognized by TLR4 and activate the MAPKs (ERK, JNK and p38 MAPK) in the process of CFP-mediated activation in macrophages, of which the effect of CFP-2 on the signal pathway immunostimulatory is speculated, and displayed in Figure 10.

## 3. Materials and Methods

### 3.1. Materials

RAW264.7 cell line was purchased from the Type Culture Collection of Chinese Academy of Sciences (Shanghai, China). DMEM, with the supplement of 100 IU mL^−1^ benzylpenicillin, 100 IU mL^−1^ streptomycin and 10% fetal bovine serum, was procured from Gibco (Fort Worth, TX, USA).

Assay kits for IL-1β, IL-6, TNF-α, pyridine and lipopolysaccharides (LPS) were acquired from Sigma Chemical Co. (Saint Louis, MO, USA). MTT and NO assay kit was obtained from Nanjing Jian cheng Bioengineering Institute (Nanjing, China). Assay kits for the quantification of messenger RNA(mRNA) were purchased from Beijing Solarbio Science & Technology Co., LtD (Beijing, China). TAK-242 (TLR4 inhibitor), SP600125 (JNK inhibitor), U0126 (ERK inhibitor) and SB203580 (P38 inhibitor) were acquired from Abmole (Houston, TX, USA). Water used in the study was produced by Milli-Q system (Millipore, Bedford, MA, USA). All other chemicals and solvents were of analytical reagent grade.

### 3.2. Isolation and Purification of Polysaccharides from Chlamys farreri

#### 3.2.1. Optimization of Extraction Conditions

The extraction conditions were optimized using three parameters, including volume ratio (30, 40, 50, 60 and 70 mL/g), extraction time (1, 2, 3, 4 and 5 h) and temperature (35, 50, 65, 80, 95 °C). The experiment was designed such that only one variable was altered at a time, while the others were kept constant. The total sugar contents were measured by phenol-sulfuric acid method, using d-glucose as a standard.

#### 3.2.2. Removal of Proteins from Crude Polysaccharides

The crude polysaccharide was deproteinized by sevag method [51] and aqueous two-phase system (ATPS) method composed of ethanol and ammonium sulfate [4,24].

##### Sevag Method

Crude polysaccharides (50 mg) were dissolved in distilled water (50 mL). It was then mixed with 20 mL Sevag reagent (1-butanol: chloroform = 1:4), and the mixture was stirred to homogenize. After 30 min, the mixture was centrifuged at 4000rpm for 10min. The supernatant, hence obtained, was collected, and the process was repeated five times. The recovery rate of polysaccharide (Rps, %) and residue rate of protein (Rpro, %) were the concentration of the polysaccharides and proteins deproteinized by sevag method to the amount added, which were calculated as the following equations:Rps (or Rpro) = C × V/m × 100%(1)
where, C: concentration of polysaccharide or protein in the deproteinized solution by sevag method; V: volumes of the deproteinized solution by sevag method; m: the detected amount of the crude polysaccharides or proteins initial dissolved. The loss rate of polysaccharide (Lps, %) was calculated as the following equations:Lps = (1 – Cps × V / m) × 100%(2)
where, Cps: concentrations of the polysaccharide in the deproteinized solution by sevag method; V: volumes of the deproteinized solution by sevag method; m: the detected amount of the crude polysaccharides initial dissolved.

##### Ethanol-Ammonium Sulfate ATPS

Crude polysaccharide (50 mg) was mixed with 80g ATPS, which was comprised of 17.7% (w/w) ethanol and 27.3% (w/w) (NH4)2SO4 at TLL (tie line length) of 35. The extraction ATPS mixture was shaken for 30 min using an electric mixer, centrifuged at 3000 rpm for 1 min. The top phase and bottom phase were isolated by a pipette and analyzed, respectively. The recovery rate of polysaccharide (Rps, %) and residue rate of protein (Rpro, %) were the concentration of the polysaccharides and the proteins deproteinized, in both the top phase and bottom phase by ATPS method, which were calculated as the following equations:Rps (or Rpro) = (Ct × Vt + Cb × Vb)/m × 100%(3)
where, Ct and Cb: concentrations of polysaccharide or protein in the top and bottom phases; Vt and Vb: volumes of the top and bottom phases; m: the detected amount of the crude polysaccharides or proteins initial dissolved. The loss rate of polysaccharide (Lps, %) was calculated as the following equations:Lps = (1 − Rps) × 100%(4)
where, Rps: recovery rate of polysaccharide by ATPS method.

#### 3.2.3. The Separation of Polysaccharide from *Chlamys farreri*

##### DEAE Cellulose-52 Column Chromatography

Crude CFP (40 mg) dissolved in 4 mL distilled water was filtered through 0.45 μm membrane and passed through a DEAE-52 cellulose column (1.6 × 20 cm), which was pre-equilibrated with distilled water. The column was eluted with distilled water at 1.0mL/min, and afterwards with a liner gradient of NaCl (0 to 1mol/L). Each fraction (5 mL eluent) was collected using an automatic collector, and detected by the phenol-sulfuric acid method. The tube number was plotted against the absorbance shown by the eluent to obtain an elution curve. Eluents with the same peak, according to the elution curve, were combined and dialyzed against distilled water for 48 h at the room temperature using the dialysis bags (molecular weight (Mw) cut off 3.5 KDa). This was followed by the concentration of the polysaccharide at 50 °C, and its subsequent freeze-drying.

##### Gel Permeation Chromatography on Sephadex G75

Each polysaccharide fraction (20 mg each) obtained after DEAE cellulose-52 column chromatography was dissolved in 2 mL distilled water, and the resultant solution was filtered through a membrane (0.45 μm). The filtrate was, afterwards, loaded onto a Sephadex G75 column (1.6 × 20 cm). The column was eluted with distilled water (0.25 mL/min), and 2.5 mL eluents (each comprising a distinct polysaccharide fraction) were collected using an automatic collector and detected by the phenol-sulfuric acid method. The elution curve was drawn by plotting the tube number against the absorbance of the eluent. The main fraction collected after gel filtration was concentrated at 50 °C, and then subjected to freeze-drying. 

### 3.3. Chemical-Physical Properties of Polysaccharides

#### 3.3.1. Determination of Glucuronic Acid Content

The glucuronic acid content of CFP was determined following the m-hydroxydiphenyl method, with glucuronic acid as a standard [19,52]. Disodium tetraborate sulfuric acid solution (1.5 mL), in a concentration of 12.5 mmol/L, was added to 0.25 mL polysaccharide solution (1 mg/mL), and was mixed thoroughly. After reacting at 100 °C for 10 min, the reaction was stopped, by transferring the reaction mixture to an ice bath. Finally, 25 μL of m hudroxydiphenyl NaOH solution (0.15%) was added to it, and the resultant solution was kept at ambient temperature for 40 min, before measuring its absorbance at 523 nm.

#### 3.3.2. Determination of Homogeneity and Molecular Weight

The molecular weight of CFP was determined through high performance gel filtration chromatography technique (HPGFC), using an HPLC system that was equipped with a TSK-gel G4000PW column (7.8 mm × 300 mm) and an evaporative light scattering detector (Alltech ELSD6000, New Westminster, BC, USA). The polysaccharide solution (2 mg/mL) was dissolved in 50 mmol/L ammonium acetate. After filtration through a 0.45 μm of membrane, the 10 μL filtrate was injected into the HPLC system. The column was eluted with 50 mmol/L ammonium acetate, at a flow rate of 0.5 mL/min. Dextran standards (80 kDa, 40 kDa, 20 kDa, 10 kDa, 5 kDa) for GPC were used, and a calibration curve was drawn.

#### 3.3.3. UV and FT-IR Analyses

The UV-vis absorption spectra of CFPs were recorded using UV-1800PC, in the wavelength range of 200–600 nm. Similarly, the FT-IR (MAGNA-IR 750, Thermo Nicolet Co., Madison, WI, USA) spectrum was determined in the frequency range of 4000–400 cm^−1^, by pressing CFPs (1 mg) and KBr (100 mg) into a pellet.

#### 3.3.4. Congo Red Analysis

The spatial conformation of CFPs was determined according to Congo red method reported in our previous publication24. CFPs (2 mg), dissolved in 1mL distilled water, were mixed with 2 mL of 100 μM Cong-red solution. The maximum absorption wavelength of the solution was measured using UV-1800 PC spectrophotometer at different NaOH concentrations (0, 0.1, 0.2, 0.3, 0.4, 0.5, 0.6, 0.7 mol/L). Curdlan was used as a positive standard in this study.

#### 3.3.5. Analysis of Monosaccharide Composition

HPLC was used to ascertain the monosaccharide composition of CFPs, following a modified procedure laid down in an earlier report [53]. Briefly, 5 mg CFP was hydrolyzed with 1 mL TFA (2 mol/L), at 120 °C for 4 h in a sealed tube. After removing excess TFA in a stream of N2, the residue was re-dissolved in 1mL distilled water. The hydrolyzed product (100 μL) was reacted with 0.6 mol/L NaOH (100 μL) and 0.5 mol/L PMP (200 μL), at 70 °C for 2 h. Finally, the PMP derivatives were extracted using CH2Cl2, and analyzed using HPLC, equipped with a UV-detector and a Hypersil ODS2 column (5 μm; 250 mm × 4.6 mm). The mobile phase comprised of 0.1 mol/L phosphate buffer (pH 6.8) and acetonitrile in a ratio of 84:16 (v/v, %), and the flow rate was maintained at 0.8 mL/min. The temperature inside the column was 30 °C, and the analytes were detected at 254 nm.

#### 3.3.6. Methylation Analysis

The pattern of glycosidic linkages present in CFPs was elucidated following methylation analysis, as described earlier [54]. After dissolving 5 mg CFP in 5 mL DMSO in the presence of N2, the resultant solution was supplemented with NaOH (20 mg) and CH3I (300 μL). The reaction mixture was then kept in the dark for 4 h, and the reaction was subsequently terminated by the addition of 2 mL distilled water. It was afterwards extracted with dichloromethane (4 mL), and the dichloromethane layer was resuspended in 1 mL water, before drying it at 50 °C under vacuum. A complete methylation was confirmed by the disappearance of O-H absorption (3200-3700 cm^−1^) in IR spectrum. Subsequently, the residue was hydrolyzed with TFA for 4 h at 120 °C. The hydrolyzed product was also dried under vacuum at 50 °C, while the residue was later reduced with NaBH4, neutralized with acetic acid and acetylated with acetic anhydride, to obtain a mixture of partially O-methylated alditol acetates.

The acetylated product was analyzed by GC-MS (QP-2010ULTRA, Shimadzu, Japan), equipped with an Agilent DB-225 ms capillary column (0.25 mm × 30 m × 0.25 µm). The injector and detector temperatures were maintained at 250 °C and 280 °C, respectively, while the temperature inside the column was 100 °C. Helium was used as the carrier gas at a flow rate of 1 mL/min. The partially methylated alditol acetates were identified by their retention time and electron ionization spectrum. The degree of branching value (DB) was obtained by using the following equation: DB = (NT + NB) / (NT + NB + NL)(5)
where NT, NB and NL represent the number of terminal, branched and linear residues, respectively.

### 3.4. Immunostimulatory Activity of CFPs

#### 3.4.1. Cell Culture

RAW264.7 cells were cultured in DMEM, supplemented with 10% (V/V) inactivated FBS, glutamine (2.0 mmol/L), penicillin (100.0 U/mL) and streptomycin (100.0 μg/mL). The cells were cultivated at 37 °C in a cell incubator, with humidified air containing 5% CO_2_, and were afterwards collected in the logarithmic phase by gentle scraping. They were then resuspended in DMEM medium for further experimentation.

#### 3.4.2. Determination of Proliferation of RAW264.7 Cells

The MTT method was used to measure the effects of CFPs on the proliferation of RAW264.7 cells. The cell suspension (1 × 10^5^ cells/mL) was seeded into a 96-well flat-bottom plate (100 μL per well). After an incubation period of 24 h, the culture supernatants were discarded, and the pre-selected doses of CFP (0, 125, 250, 500, 1000 and 2000 μg/mL) and LPS (5 μg/mL), which were proven to have no obvious inhibitory effect on cell growth, were added into the wells. An incubation period of 24 or 48 h was implemented for the cultured cells in the presence of 5% CO2. The cells were subsequently treated with MTT in the dark for 4 h, followed by the removal of the medium. Formazan crystals, hence obtained, were suspended in 200 μL/well DMSO. Cell viability was determined as a ratio of sample absorbance and control absorbance at 570 nm.

#### 3.4.3. Macrophages Phagocytosis Assay

The neutral red uptake assay was used to measure the efficacy of phagocytosis of RAW264.7 cells [55]. Each well of a 96-well flat-bottom plate was seeded with 200 μL of the cell suspension (1 × 10^5^ cells/mL). The culture supernatants were removed after incubating the cells for 24 h. This was followed by the addition of CFP (0, 125, 250, 500, 1000 and 2000 μg/mL) and LPS (5 μg/mL) in the culture wells, and incubation at 37 °C in 5% CO2 for 24 or 48 h. Following the removal of the supernatant, each well was supplemented with 100 μL neutral red solution (0.1%). The cells were subjected to another incubation for 30 min, followed by the removal of the supernatant and rinsing of the cells in PBS. The cells were subsequently incubated with 100 μL cell lysis buffer (ethanol and glacial acetic acid in a ratio of 1:1) for 2 h, and the absorbance was measured at 540 nm. The inhibitory ratio of CFPs against RAW264.7 cell proliferation was calculated by the following formula: Inhibitory ratio (%) = [1 − (Asample − Ablank) / (Acontrol − Ablank)] × 100%(6)
where Acontrol and Ablank were the absorbance of the system without addition of CFPs and RAW264.7 cells, respectively.

#### 3.4.4. Measurement of NO Production

The production of NO was measured in Griess assay by seeding the cell suspension (1 × 10^5^ cells/mL) in a 96-well flat-bottom plate (200 μL per well), and incubating it for 24 h. After discarding the supernatant, the culture wells were supplemented with CFP (0, 125, 250, 500, 1000 and 2000 μg/mL) and LPS (5 μg/mL). This was followed by incubation for 24 h or 48 h at 37 °C in the presence of 5% CO2, and the collection of the supernatant. The supernatant (50 μL) obtained after centrifugation (4000 rpm; 5 min) was treated in dark with an equal volume of Griess reagent at ambient temperature, and the absorbance was calculated at 540 nm.

#### 3.4.5. Measurement of TNF-α, IL-1β, IL-6 and IL-10

The concentrations of TNF-α, IL-1β and IL-6 were assessed using ELISA kits, according to the manufacturer’s instructions.

#### 3.4.6. Quantification of Messenger RNA (mRNA) (Bejing Solarbio Science & Technology Co., LtD. Beijing, China)

The manufacturer’s protocol was followed for utilizing the total RNA extraction kit, in order to extract RNA from 24 h CFP and LPS treated RAW264.7 cells. The extracted RNA was subsequently converted into cDNA using reverse transcription. Relative target gene quantification was conducted on the real-time PCR system using SYBR green. PCR reactions were performed for 5 min at 95 °C, and then 40 cycles of 10 s at 95 °C and 30 s at 60 °C. Each reaction was performed in triplicate. The expression of each gene was compared with GAPDH expression that served as a control. The relative expressions of mRNAs were calculated using the comparative 2-∆∆ct method, and were normalized using GAPDH. The nucleotide sequences of the primers used were GADPH (forward, 5-GGTGAAGGTCGGTGTGAACG-3; reverse, 5-CTCG CTCCTGGAAGATGGTG-3); iNOS (forward, 5-CGGCAAACATGACTTCAGGC-3; reverse, 5-CTCGCTCCTGGAAGATGGTG-3); IL-6 (forward, 5-TACTCGGCAAA CCTAGTGCG-3; reverse, 5-GTGTCCCAACATTCATATTGTCAGT-3); TNF-α (forward, 5-AGATAGCAAATCGGCTGACG-3; reverse, 5-ACGGCATGGATCTCA AAGAC-3); and IL-1β (forward, 5-TCTTTTGGGGTCCGTCAACT-3; reverse, 5-GC AACTGTTCCTGAACTCAACT-3).

#### 3.4.7. Related Signal Path Experiment

RAW264.7 cells were pretreated with TLR4 (2.5 μmol/L), SP600125 (10 μmol/L), SB203508 (10 μmol/L) and U0126 (10 μmol/L); the culture supernatants were removed after incubating the cells for 1 h. Then, the cells were incubated with CFP-2 (500 μg/mL) for an additional 24 h. The concentrations of NO, TNF-α, and L-6 were assessed according to the manufacturer’s instructions.

### 3.5. Statistical Analyses

Each experiment was performed in triplicate to minimize deviation. The data, hence obtained, were presented as mean ± SD and subjected to variance (ANOVA) analysis, where *p* < 0.05 was assumed to be statistically significant. The statistical analyses were performed using the Statistical Package for the SPSS Statistics 20 software (IBM Co., Armonk, NY, USA).

## 4. Conclusions

Two polysaccharides (CFP-1 and CFP-2) were extracted from *Chlamys Farreri*, which were deproteinized by ethanol-ammonium sulfate ATPS, and separated by DEAE collulose-52 column chromatography and Sephadex G-75. Furthermore, their physicochemical properties were elucidated, including molecular weight, monosaccharide composition, the glucuronic acid content, UV and IR analyses and glycosidic linkage determination. The results showed that CFP-1 and CFP-2 were mainly composed of glucose with different molecular weight, glucuronic acid content and branching degree. Besides, they lack the triple-helix structure, as elucidated in Cong-red experiment. Both crude CFP and CFP-2 were found capable of significantly enhancing the phagocytic activity, and were able to increase the production of the cytokines (NO, TNF-α, IL-1β and IL-6), by activating iNOS, IL-6 and TNF-α gene expressions in RAW264.7 cells. CFP can be accepted by TLR4 and activate the MAPKs (ERK, JNK and p38 MAPK) in the process of CFP-mediated activation in macrophages. CFP-1, however, failed to produce desired effects in this regard. A comparison of the immunostimulatory activities exhibited by CFP-1 and CFP-2 and their physicochemical properties pointed towards the evident role of molecular weight, glucuronic acid content and branching degree, in determining the immunostimulatory activity of polysaccharides.

## Figures and Tables

**Figure 1 marinedrugs-18-00429-f001:**
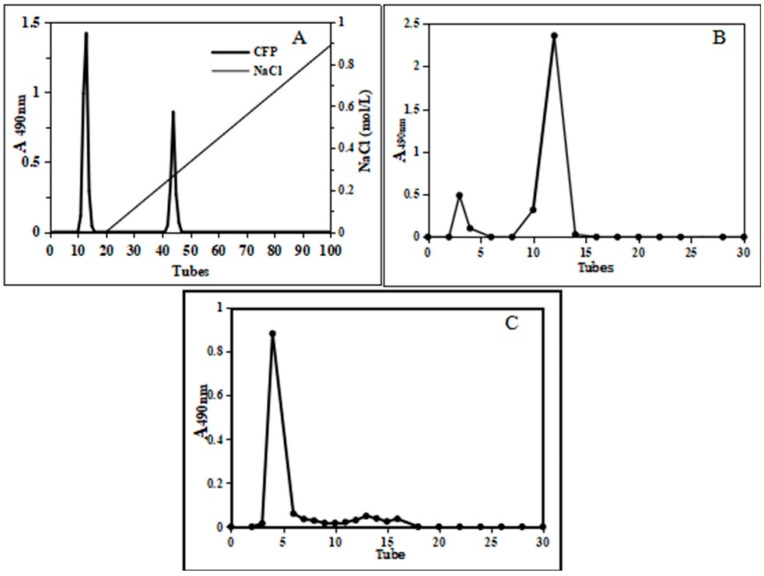
Various column chromatography elution curve of CFP. (**A**): DEAE cellulose-52 column chromatography of crude CFP, (**B**): Sephadex G75 column chromatography of neutral polysaccharide, (**C**): Sephadex G75 column chromatography of acidic polysaccharide

**Figure 2 marinedrugs-18-00429-f002:**
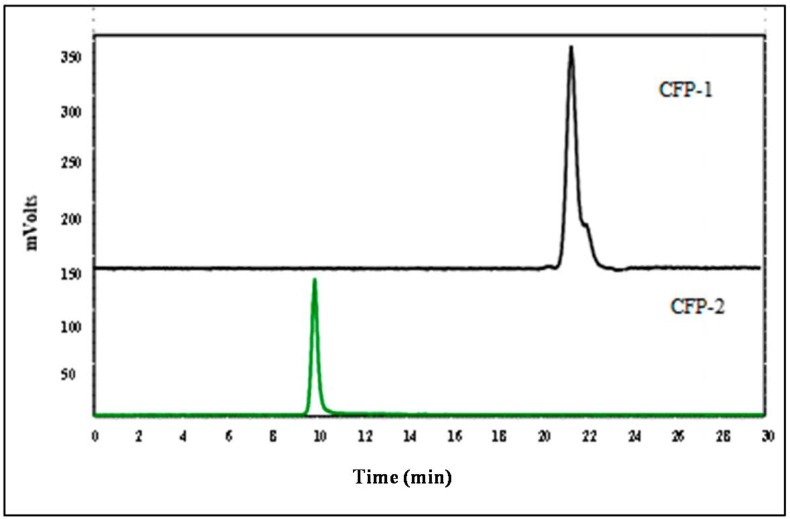
HPLC elution of CFP-1 and CFP-2.

**Figure 3 marinedrugs-18-00429-f003:**
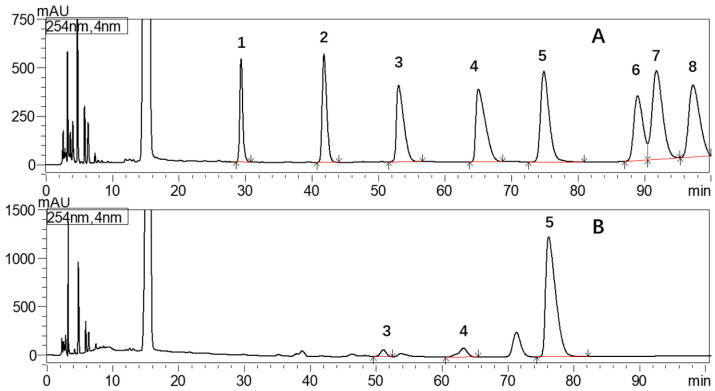
Monosaccharides through HPLC. (**A**) Standard monosaccharide: 1. Mannose; 2. Rhamnose; 3. Glucuronic acid; 4. Galacturonic acid; 5. Glucose; 6. Galactose; 7. Arabinose; 8. Fucose. (**B**) Monosaccharide of crude CFP.

**Figure 4 marinedrugs-18-00429-f004:**
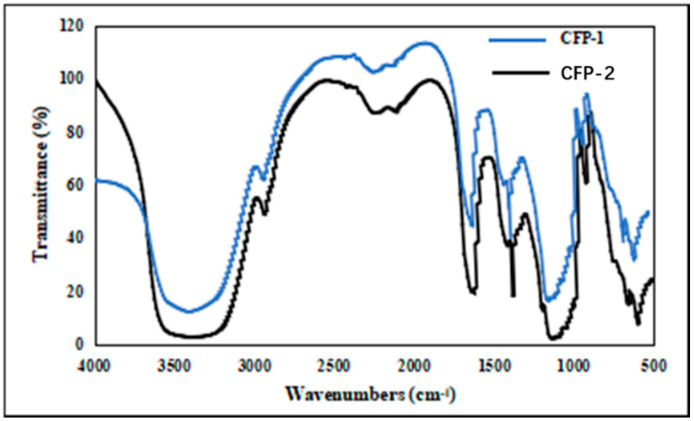
Fourier transform infrared (FT-IR) spectrum of CFP-1and CFP-2 in a KBr pellet over the range of 400–4000 cm^−1^.

**Figure 5 marinedrugs-18-00429-f005:**
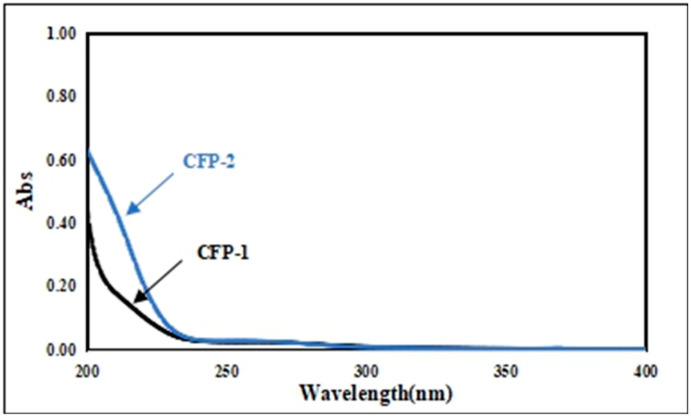
UV scanning spectrum of CFP-1 and CFP-2.

**Figure 6 marinedrugs-18-00429-f006:**
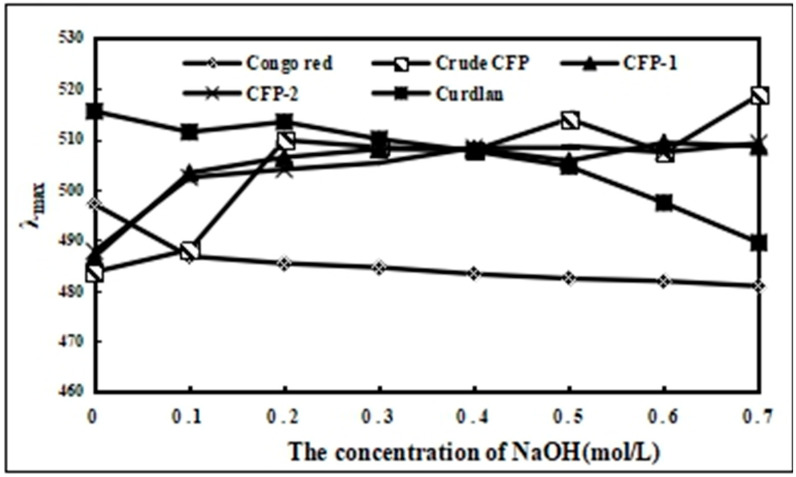
The Congo -Red experiment of crude CFP, CFP-1 and CFP-2.

**Figure 7 marinedrugs-18-00429-f007:**
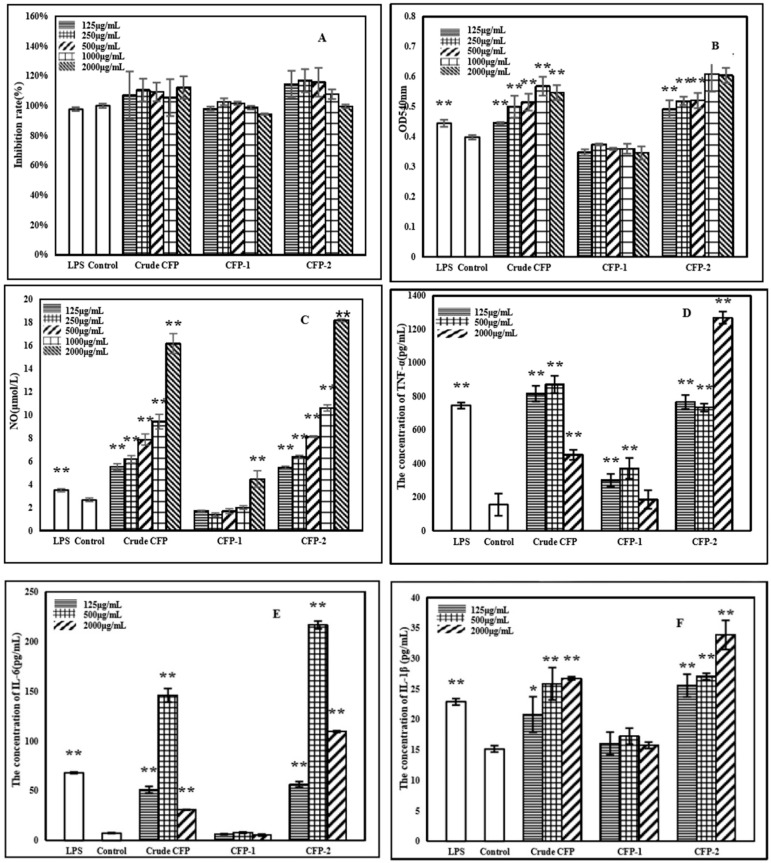
Effects of polysaccharides on the viability of RAW264.7 cells (**A**); Effects of polysaccharides on the phagocytosis activity of RAW264.7 cells (**B**); Effects of polysaccharides on the production of NO (**C**), TNF-α (**D**), IL-6 (**E**) and IL-1β (**F**) of RAW264.7. The group without polysaccharide was used as the normal control, and LPS (5 μg/mL) was used as the positive control group. The data shown are means ± SD (*n* = 3). All data were analyzed statistically using a one-way analysis of variance. (*****) *p* < 0.05 and (******) *p* < 0.01, compared with the normal control, respectively.

**Figure 8 marinedrugs-18-00429-f008:**
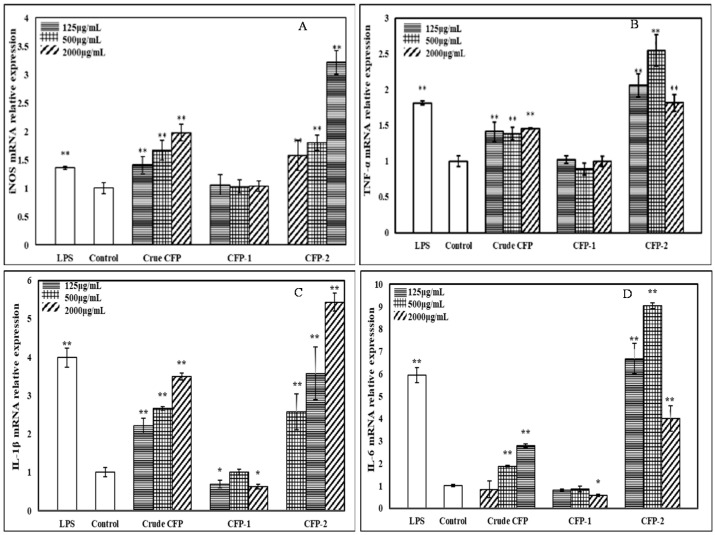
Effects of CFPs on iNOS (**A**), TNF-α (**B**), IL-1β (**C**) and IL-6 (**D**) mRNA expression. The group without polysaccharide was used as the normal control, and LPS (5 μg/mL) was used as the positive control group. The data shown are means ± SD (*n* = 3). All data were analyzed statistically using one-way analysis of variance. (*) *p* < 0.05 and (**) *p* < 0.01 compared with the normal control, respectively.

**Figure 9 marinedrugs-18-00429-f009:**
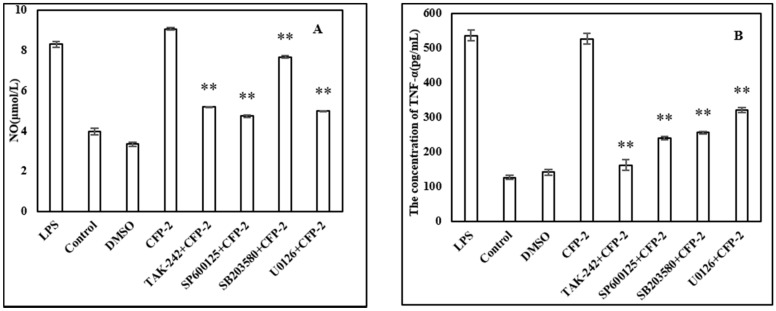
Effect of inhibitors on NO production in CFP-activated RAW264.7 cells (**A**); Effect of inhibitors on TNF-α production in CFP-activated RAW264.7 cells (**B**). The data shown are means ± SD (*n* = 3). All data were analyzed statistically using one-way analysis of variance. (*****) *p* < 0.05 and (******) *p* < 0.01 compared with the CFP-2 group, respectively.

**Figure 10 marinedrugs-18-00429-f010:**
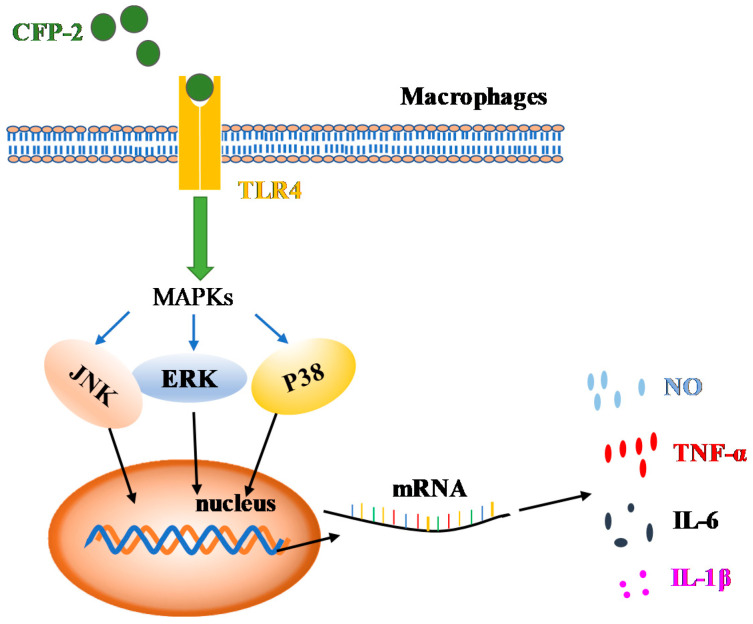
Illustration of immune system activation by CFP.

**Table 1 marinedrugs-18-00429-t001:** Effect of solvent to material (V/W ratio), extraction temperature and extraction time on the total sugar yield. Values are represented as the means ± SD, as determined from triplicate experiments.

Factors	Yield of Total Sugar (%)				
V/W ratio (65 °C, 3 h)	30	40	50	60	70
	26.31 ± 0.82	26.95 ± 0.60	28.27 ± 1.12	28.78 ± 0.64	28.50 ± 0.73
Temperature (°C) (50 V/W, 3 h)	35	50	65	80	95
	21.84 ± 0.47	22.93 ± 0.52	25.21 ± 0.70	25.64 ± 1.12	25.74 ± 0.33
Time (h) (50 V/W, 65 °C)	1	2	3	4	5
	24.04 ± 0.22	25.10 ± 0.34	25.47 ± 0.31	26.30 ± 0.11	26.45 ± 0.14

**Table 2 marinedrugs-18-00429-t002:** Comparison of two methods for de-proteninzation of crude CFP. Values are represented as the means ± SD as determined from triplicate experiments.

Method		R_ps_ (%) ^a^	L_ps_ (%) ^a^	R_pro_ (%) ^a^
ATPS	Top phase	19.34 ± 0.62	14.83 ± 2.77	16.60 ± 0.80
	Bottom phase	65.82 ± 2.16		8.04 ± 0.45
Sevag		81.72 ± 2.37	18.28 ± 2.37	39.81 ± 4.10

^a^ R_ps_ denotes recovery ratio of polysaccharide, L_ps_ denotes loss ratio of polysaccharide, and R_pro_ denotes residue ratio of protein.

**Table 3 marinedrugs-18-00429-t003:** GC-MS of alditol acetate derivatives from the methylated product of the CFP-1 and CFP-2.

Retention Time (min)	Methylated Sugar	Linkage Types	Molar Ratio (%)
CFP-1			
13.80	2,4,6-Me3-O-methyl-d-Glcp	2-d-Glcp	2.12
13.98	2,3,6-Me3-Omethyl-d-Glcp	4-d-Glcp	1.00
15.68	1,3,5,6-Me4-O-methyl-d-Glcp	3,4,6-d-Glcp	6.54
CFP-2			
12.92	2,3,4,6-Me4-O-methyl-d-Glcp	T-d-Glcp	1.00
13.98	2,3,6-Me3-O-methyl-d-Glcp	4-d-Glcp	9.73
15.07	2,6-Me2-O-methyl-d-Glcp	3,4-Glcp	1.72

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
