# Peer review of "Comparison of Physicochemical Characteristics and Macrophage Immunostimulatory Activities of Polysaccharides from Chlamys farreri"

_marinedrugs, 2020, doi:10.3390/md18080429_

Round 1
Reviewer 1 Report
General comments
The authors extracted two polysaccharides (CFP-1 and CFP-2) were from Chlamys farreri determined their structures and immunostimulatory function. They found both crude CFP and CFP-2 can enhance phagocytic activity and increase cytokines (NO, TNF-α, IL-1β and IL-6) production in RAW264.7 cells through TLR4/MAPKs pathway. In general, the experiments were well conducted and the manuscript were well-written. The author team also had several relative work published. This manuscript could be published after minor revision.
Specific comments
- The activity of CFP in this study showed only phagocytic and immunostimulatory functions but not immunomodulatory. All immunomodulatory description in this manuscript including title should be changed by immunostimulatory activity.
- The high dose of CFP used in treating RAW cells could induce cell death which may explain the decreasing cytokines production as compared in lower dose. The author should show dose relative effect of CFP on cell injury/death and discuss it.
Author Response
Reviewer 1
1.Comment:
The activity of CFP in this study showed only phagocytic and immunostimulatory functions but not immunomodulatory. All immunomodulatory description in this manuscript including title should be changed by immunostimulatory activity.
Response:
Thank reviewer for the valuable suggestion. We have changed all the immunomodulatory description into immunostimulatory activity which were set to blue.
- Comment:
The high dose of CFP used in treating RAW cells could induce cell death which may explain the decreasing cytokines production as compared in lower dose. The author should show dose relative effect of CFP on cell injury/death and discuss it.
Response:
Thank reviewer for the valuable suggestion. The doses selected in treating RAW cells were all pre-selected through experiments. The results show that high dose of CFP have no obvious inhibitory effect on cell growth,.and the effects of polysaccharides on RAW264.7 cell proliferation also indicated the nontoxicity of CFPs (≤2000 μg/mL). We have modified the relevant content in line 461 to 463, thank you very much.

Reviewer 2 Report
In this manuscript by Shi et al, the authors describe the purification and structural characterization of alpha-glucosides from the scallop Chlamys farreri. They go on to characterize their effects on the macrophage cell line RAW246.7 and their abilities to activate the macrophages to produce pro-inflammatory molecules. They then perform inhibitory studies to determine CFP-2 activates at least partially through the TLR4/MAPK pathways.
Overall the research appears logical and well-executed to describe the purification and structure of the CFP molecules, and provides a starting point investigating the possible immunomodulatory potential of these polysaccharides. Please see minor comments below:
- Line 24: "less more GAc and BD"; less or more?
- Line 76: "A slight decline in yield, nonetheless, was observed at 95C." This is not seen in Table 1 (yield increases from 25.64% to 25.74% between 80C and 90C).
- Table 2: "Rate" typically denotes something over time, yet the values reported are simply percentages.
- Figure 1: Please specify which fractions were collected for downstream characterization.
- Figure 3B: Is there possibly a rhamnose peak of similar intensity and drift compared to the standards as the glucuronic acid peak?
- Figure 6, and to a lesser extent Figs 7 and 8, are difficult to understand in their present form. I would suggest using different shades of grey, or ideally colour to differentiate the experimental groups, and limit the use of different symbols and patterns.
- Would it not make more sense to discuss Fig 8 before Fig 7, as transcription occurs before translation?
- To further explore the consequences of these immunomodulatory effects, the authors could conduct microbial killing assays with the RAW cells in the presence or absence of the CFP molecules, to determine if they enhance the antimicrobial effects Fig 7 alludes to. Such an experiment should not impede editorial acceptance of this manuscript though.
- Figure 9 would be far more informative if the targets of each inhibitor were stated. Were combinations of the inhibitors tested, to see if they had additive or synergist effects?
- Figure 9 could benefit from western blot analysis of the phosphorylation of JNK, ERK and p38, and their subsequent reduction following CFP-2 incubation. Such an experiment should not impede editorial acceptance of this manuscript though.
- Line 491: please name the manufacturer of the ELISAs.
Author Response
Reviewer 2
1.Comment:
Line 24: "less more GAc and BD"; less or more?
Response:
Thank you for careful suggestion. We have modified the relevant content(line24).
2.Comment:
Line 76: "A slight decline in yield, nonetheless, was observed at 95C." This is not seen in Table 1 (yield increases from 25.64% to 25.74% between 80C and 90C).
Response:
Thank you for careful suggestion, it was a writing error, We have modified the relevant content in line76.
3.Comment:
Table 2: "Rate" typically denotes something over time, yet the values reported are simply percentages.
Response:
Thank reviewer for the careful suggestion. We have changed "rate" by "ratio" in Table 2, We thought it would be more appropriate.
4.Comment:
Figure 1: Please specify which fractions were collected for downstream characterization.
Response:
Thank you for careful suggestion. We have modified the content in line103 to104 which were set to bule.
5.Comment:
Figure 3B: Is there possibly a rhamnose peak of similar intensity and drift compared to the standards as the glucuronic acid peak?
Response:
Thank you for your considerable comment. We have also considered whether the small peak in front of peak 3 was rhamnose. We have analyzed the standard of rhamnose and found that the retention time of the standard sample and the retention time of the sample were not within the allowable range, and the intensity and drift were greater than the nearby peak of the glucuronic acid.
6.Comment:
Figure 6, and to a lesser extent Figs 7 and 8, are difficult to understand in their present form. I would suggest using different shades of grey, or ideally colour to differentiate the experimental groups, and limit the use of different symbols and patterns.Would it not make more sense to discuss Fig 8 before Fig 7, as transcription occurs before translation?
Response:
Thank reviewer for the valuable suggestion. We thought Figs 7 and 8 were a little bit complicated and required readers to take times to understand, but it clear enough to differentiate the experimental groups. For the second question, in theory, we thought we should first discover the changes in various cytokines, and then explore the changes in its mRNA levels, thank you very much.
7.Comment:
To further explore the consequences of these immunomodulatory effects, the authors could conduct microbial killing assays with the RAW cells in the presence or absence of the CFP molecules, to determine if they enhance the antimicrobial effects Fig 7 alludes to. Such an experiment should not impede editorial acceptance of this manuscript though.
Response:
Thank reviewer for the valuable suggestion. In deed, if we conducted the microbial killing assays with the RAW cells test could more intuitively show the enhancement effect of CFP on the immunostimulatory activity, we were very sorry that this text did not involve in this part of the experiment, but ours study was focuses on the effect of CFP on the phagocytosis of macrophages and the secretion of cytokines, which was proved that CFP has certain immunostimulatory activity.
8.Comment:
Figure 9 would be far more informative if the targets of each inhibitor were stated. Were combinations of the inhibitors tested, to see if they had additive or synergist effects? Figure 9 could benefit from western blot analysis of the phosphorylation of JNK, ERK and p38, and their subsequent reduction following CFP-2 incubation. Such an experiment should not impede editorial acceptance of this manuscript though.
Response:
Thank reviewer for the valuable suggestion. We were very sorry that the research on signal pathways in this paper had not been fully studied. We will enrich the related experiment in our future research, thank you very much.
9.Comment:
Line 491: please name the manufacturer of the ELISAs.
Response:
Thank reviewer for the careful suggestion. We have modified the relevant content which was set to blue in line326-368.
